# Overview of Nucleocapsid-Targeting Vaccines against COVID-19

**DOI:** 10.3390/vaccines11121810

**Published:** 2023-12-03

**Authors:** Alexandra Rak, Irina Isakova-Sivak, Larisa Rudenko

**Affiliations:** Department of Virology, Institute of Experimental Medicine, St. Petersburg 197022, Russia; isakova.sivak@iemspb.ru (I.I.-S.); vaccine@mail.ru (L.R.)

**Keywords:** SARS-CoV-2, nucleocapsid protein, COVID-19 vaccine, cross protection, recombinant protein, N protein evolution

## Abstract

The new SARS-CoV-2 coronavirus, which emerged in late 2019, is a highly variable causative agent of COVID-19, a contagious respiratory disease with potentially severe complications. Vaccination is considered the most effective measure to prevent the spread and complications of this infection. Spike (S) protein-based vaccines were very successful in preventing COVID-19 caused by the ancestral SARS-CoV-2 strain; however, their efficacy was significantly reduced when coronavirus variants antigenically different from the original strain emerged in circulation. This is due to the high variability of this major viral antigen caused by escape from the immunity caused by the infection or vaccination with spike-targeting vaccines. The nucleocapsid protein (N) is a much more conserved SARS-CoV-2 antigen than the spike protein and has therefore attracted the attention of scientists as a promising target for broad-spectrum vaccine development. Here, we summarized the current data on various N-based COVID-19 vaccines that have been tested in animal challenge models or clinical trials. Despite the high conservatism of the N protein, escape mutations gradually occurring in the N sequence can affect its protective properties. During the three years of the pandemic, at least 12 mutations have arisen in the N sequence, affecting more than 40 known immunogenic T-cell epitopes, so the antigenicity of the N protein of recent SARS-CoV-2 variants may be altered. This fact should be taken into account as a limitation in the development of cross-reactive vaccines based on N-protein.

## 1. Introduction

To date, more than 695 million cases of SARS-CoV-2 infection have been recorded worldwide, and this number continues to increase with the emergence of new viral variants [1]. Despite the end of the pandemic, cases of deadly COVID-19 infection continue to be reported as a result of successive waves of worldwide spread of new SARS-CoV-2 variants, which is associated with the high antigenic variability of the virus [2]. Immunocompromised individuals are another reservoir for selection and accumulation of escape mutations in the SARS-CoV-2 genome [3]. The emergence of new SARS-CoV-2 variants as a result of long-term virus persistence in patients with acute lymphoblastic leukemia [4], B-cell leukemia complicated by type 2 diabetes mellitus [5], acute myeloid leukemia and spleen lymphoma [6] has been reported.

The most effective measure of COVID-19 prevention is vaccination [7], and most licensed COVID-19 vaccines use the SARS-CoV-2 surface spike (S) protein as a target antigen, which seems appropriate since it is exposed on the virion surface and mediates viral entry into cells [8]. However, the extremely high mutational rate of this antigen renders vaccines ineffective against infections caused by new SARS-CoV-2 variants [9]. In this regard, more effective vaccination strategies should target not only the variable S protein but also conserved viral antigens such as the nucleocapsid (N) protein [10]. Notably, during COVID-19 progression, the formation of antiviral antibodies is observed not only in the surface coronavirus antigens but also in the N protein, which is abundantly produced in the cytoplasm of infected cells [11] and is also detected on their surface [12]. This viral component is involved in the fundamental processes of the SARS-CoV-2 life cycle: viral ssRNA replication, virion assembly and viral genome encapsidation [13]. In general, the N protein performs at least three functions in pathogen virulence: it ensures fitting and packaging of the viral genome, enhances viral mRNA transcription by condensation of genomic RNA in the cell and suppresses host immune responses, in particular interferon mediated [14].

However, despite the rather high degree of N conservation compared to the other surface antigens of SARS-CoV-2 [15], it should be noted that adaptive mutations slowly appear in this antigen during viral evolution [16]. Their influence on N antigenic variability and adaptability, i.e., on the formation of the virus antigenic repertoire as a result of evolution under the pressure of the host immune response, has not yet been explored. At the same time, the study of changes in the immunogenic and antigenic properties of the SARS-CoV-2 N protein under the influence of such escape mutations seems to be very relevant both for assessing the performance of existing N-based diagnostic test systems and for developing cross-protective vaccines against COVID-19. A number of vaccine candidates have been developed based on recombinant protein, viral vectors or mRNA technologies, implying different mechanisms of action. Although the main mechanism of N-based vaccine performance is the induction of T-cell immune responses [10], their administration inevitably stimulates the formation of N-specific antibodies, and it is very important to study their protective properties and autoimmune activity to assess the effectiveness and safety of vaccination.

Anti-N antibodies produced in response to infection with different strains of SARS-CoV-2 have been shown to have a longer circulation time than anti-S IgG [17] and a high degree of cross-reactivity [18]. Although antibodies against the N protein do not have neutralizing activity [19], they are intensively produced during infection [18] and may provide antiviral protection, serving as activators of complement cascade reactions [20] or antibody-dependent cellular phagocytosis/cytotoxicity (ADCP/ADCC) [21]. In addition to natural infection, the production of N-specific antibodies can be stimulated by immunization with whole-virion inactivated [22,23] or live attenuated [24] vaccines; however, the mechanism of involvement of these immunoglobulins in protection against COVID-19 remains largely unstudied. Moreover, there is evidence that anti-N antibodies may have autoimmune properties, provoking the development of various immunopathological conditions [25].

In this review, we summarized the current evidence for the development of N-based COVID-19 vaccines and discussed how slow evolutionary changes in N sequences may affect the performance of such vaccines in terms of inducing cross-reactive B- and T-cell responses. The aim of the review was to investigate the feasibility of developing N protein-based vaccines under conditions of constant virus evolution, as well as to assess the variability of the N antigen itself and its possible impact on the antigenic properties of N-based vaccines.

## 2. SARS-CoV-2 Nucleoprotein Structure and Functions

The SARS-CoV-2 N protein is a hydrophobic positively charged molecule encoded by the 9th ORF and consisting of 419 amino acid residues. It shares 91% and 49% homology with the N proteins of SARS-CoV and MERS-CoV, respectively, and contains nuclear localization signals. Several phosphorylation sites have been predicted as points of regulation of the N protein functions by protein kinases [26].

The N protein has a modular organization, which means that intrinsically disordered regions (IDRs) and conserved structural domains can be distinguished in the N sequence [27]. The first group includes the flanking N- and C-tails and a central flexible Ser/Arg-rich linker region that interrupts two conserved structural regions: the N-terminal domain (NTD) and the C-terminal domain (CTD) [14]. The NTD (residues 44–174) contains positively charged amino acids, and the linker includes a serine/arginine-rich region (residues 175–203), so they are involved in the direct interaction with the viral genome [28,29] (Figure 1).

The N protein has the ability to self-assemble into oligomers and helical filaments, which provides efficient RNP assembly [32]. Importantly, CTD, which also facilitates RNA binding, is able to self-assemble into conglomerates and mediate N tetramerization [33]. The N protein polymerization is believed to occur as a result of complete positive charge reduction via phosphorylation of Ser/Arg residues in the linker region [34].

A unique property of the N protein of SARS-CoV-2 and some other viruses, such as HIV-1, is the ability to separate liquid–liquid phases due to electrostatic interactions [35]. The resulting biomolecular condensates, including the complexed N protein molecules and viral genomic RNA, can be formed both in vitro and in living cells [36]. It is still unclear which domains are involved in this process, but it is widely accepted that it is regulated by the phosphorylation of the linker region [37] and by the acetylation of some amino acid residues in the dimerization domain [38]. Phase separation is thought to be induced by RNA [37] and M protein [39], and the latter may be paired with N or serve as a coating component of N-based annular structures. The N protein condensates may inhibit stress granules in target cells via interaction with G3BP1 and thereby suppress host immunity [39]. The main but not the only function of the N protein is genome packaging and assembly of RNP particles; therefore, numerous interaction sites with nucleic acids have been identified in this molecule. One of them is considered to be a positively charged groove located between the protruding basic β-hairpin (β2′–β3′) and the core of the molecule, in particular the NTD containing the RNA-binding arginine residues R92, R107 and R149 [28]. A positively charged canyon located in the CTD and consisting of K256, K257, K261 and R262 residues has also been reported as an RNA binding motif [34,40,41,42]. The N-terminal tail and linker region are also hypothesized to be involved in RNA binding or its enhancement [43], as they have been reported to serve as affinity enhancers and allosterical regulators of RNA-N interactions [44]. In contrast, the inhibition of N interaction with genomic RNA via NTD was observed in the presence of oligomeric poly-G RNAs affecting RNP stability [45].

Apparently, an alternative function of the N protein is the suppression of host immune responses. So, it was found to be able to interfere with the pyroptosis of infected cells by inhibiting gasdermin D cleavage by caspase-1 and thereby hindering antigenic presentation. Moreover, IL-1β secretion by monocytes infected with SARS-CoV-2 was ablated [46]. A similar effect of the N protein on innate immune responses was found as a result of inhibition of Lys63-linked polyubiquitination of the mitochondrial antiviral signaling protein [38].

## 3. Development of N-Based Broadly Protective Vaccine Prototypes

The first N-based vaccines were proposed at the very beginning of the COVID-19 pandemic by several research groups already experienced in vaccine development. Using the same technologies and advances, different prototypes of universal N-based vaccines were produced and evaluated in preclinical and clinical trials.

### 3.1. Recombinant Protein Vaccines

There are multiple vaccine candidates developed by expression of the N protein in *E.coli* cells and combining purified antigens with potent adjuvants. For example, Ghaemi et al. described a combined saponin-adjuvanted vaccine developed on the basis of recombinant RBD domain and N protein of SARS-CoV-2. When administered three times subcutaneously to mice, it stimulated the formation of virus-specific neutralizing IgG antibodies, caused a marked increase in the number of CD4^+^ and CD8^+^ T cells and elicited a balanced Th1/Th2 immune response [47].

Similarly, Thura et al. [48] examined the full-length and fragmented N protein as a long-lasting universal vaccine candidate. These vaccine variants were highly immunogenic as they evoked vigorous humoral N-specific responses in mice for several months, serving as B cell activators.

The highly immunogenic vaccine OVX033 based on the full-length N (B.1) protein of SARS-CoV-2 fused to the OVX313 heptamerization domain was designed to protect against infections caused by different sarbecoviruses, using the same technology previously described for a universal influenza vaccine based on multimeric NP [49]. Vaccination with this prototype led to the generation of non-neutralizing but specific antibodies involved in Fc-mediated cellular responses. In a hamster challenge model, the OVX033 injected with a cholesterol/squalene-based adjuvant provided cross protection against B.1, B.1.617.2 and B.1.1.529 SARS-CoV-2 VOCs, as manifested by reduced weight loss, impaired pulmonary replication of the virus, and reduced lung tissue damage [50].

A combined vaccine consisting of two immunogenic spike protein regions and a full-length N antigen formulated with aluminum hydroxide with or without monophosphoryl lipid A was studied by Özcengiz et al. The authors demonstrated recognition of the recombinant proteins by the sera of COVID-19 convalescents, as well as induction of high titers of anti-P1, anti-P2 and anti-N IgG (including neutralizing antibodies) and interferon-gamma (IFN-γ) secreting T cells in mice. The developers note that the vaccine with adjuvant containing monophosphoryl lipid A is more suitable for immunization of elderly individuals, because it promotes the development of Th1 immune response, while the use of aluminum hydroxide alone is more justified for vaccination of young people [51].

Feng et al. proposed the whole recombinant N protein of SARS-CoV-2 as a universal vaccine basis, and the correct antigenicity of the N antigen was validated using the sera of COVID-19 convalescents. The authors demonstrated a correlation of anti-N antibody titers with COVID-19 severity. The immunoreactivity of the N protein was also investigated in the BALB/c model: immunized mice showed active N-specific IgG and IgM generation, as well as serum IFN-γ production [52].

In a study by Nazarian et al., three different adjuvants (Alum, AS03 and Montanide) were used for the immunization of mice, rabbits and primates with a mix of recombinant RBD, truncated spike (lacking the N-terminal domain of S1) and full-length N protein. In all cases, the authors revealed the absence of toxicity and intensive production of specific IgG antibodies, as well as cellular responses, especially when the SARS-CoV-2 antigens were adjuvanted with AS03 and Montanide. Interestingly, neutralizing antibodies were produced only when animals were immunized with Alum adjuvant [53].

A squalene-adjuvanted vaccine comprised of the recombinant N protein expressed in *E. coli* was named Convacell and tested in mice (using BALB/c and SCID mice engrafted with human PBMCs), rabbit, hamster and non-human primate (NHP) models. The vaccine was found to be safe and immunogenic, inducing the strong formation of N-specific IgG, Th1/Th2-type cytokine responses in mice and rabbits and active anti-N CD4^+^/CD8^+^ T-cell generation in marmosets. The vaccine provides protection to vaccinated hamsters by reducing lung damage, viral replication rates and weight loss upon SARS-CoV-2 challenge [54].

### 3.2. Nanoparticle-Based Vaccines

An alternative vaccine prototype based on virus-like particles carrying S, N, M and E antigens and morphologically similar to the SARS-CoV-2 virions was proposed by Yilmaz et al. The best results on induction of specific humoral and Th1-biased T-cell responses were obtained by adsorption of particles on Alum followed by mixing with CpGOND-K3 adjuvant. A challenge study on the model of K18 hACE2 transgenic mice revealed reduced viral load and lung lesions in immunized animals [55].

Unique glyconanoparticles were created by Gao et al. [56] by conjugation of whole recombinant N protein with a number of high-affinity dextran derivatives. The vaccines induced the production of anti-N antibodies in mice and rabbits and triggered potent and prolonged N-specific CTL responses against cells infected with different SARS-CoV-2 strains.

### 3.3. Nucleic Acid-Based Vaccines

To develop a cross-reactive DNA vaccine against COVID-19, Appelberg and coauthors [57] proposed a candidate combining RBD loops from huCoV-19/WH01, Alpha and Beta VOCs, as well as M and N genes. The resulting anti-spike antibodies were able to neutralize not only homologous viruses but also SARS-CoV-2 strains Delta and Omicron and elicited N-specific T-cell responses.

Despite the fact that N-expressing mRNA vaccines appeared to be immunogenic when administered alone [58], the best protection results were obtained when such vaccine prototypes were used in combination with S- or RBD-encoding mRNAs [58,59]. A potent induction of antigen-specific IgG, proinflammatory cytokines and CD4^+^/CD8^+^ T cells, accompanied by reduction of mortality rates, weight loss and lung damage after challenge with SARS-CoV-2, were observed as the major results of vaccination with combined vaccines.

### 3.4. Viral Vector-Based Vaccines

Viral vectors are designed to efficiently deliver target antigens into the cells, and it was expected that this approach would ensure proper presentation of the N antigen to the host immune system. Unexpectedly, mice infected with a vesicular stomatitis virus (VSV) vector carrying the N gene of SARS-CoV-2 exhibited reduced innate cell responses, increased viral load and high morbidity, probably as a result of N-based inhibition of immune responses [38].

In an attempt to enhance the immunogenicity of an Ad5-based vaccine carrying the S gene, Dangi et al. combined it with an Ad5-vector expressing the N protein. A marked increase in the number of CD4^+^, CD8^+^ T cells and titer of specific antibodies were found in vaccinated K18-hACE2 mice. Although the addition of the N protein to the vaccine did not provide enhanced protection of lung tissue, vaccination with a mix of two vectors significantly reduced the severity of virus-induced brain lesions [60].

A similar vaccine based on the human adenovirus type 5 vector carrying the N gene instead of spike, showed its protective efficacy in preclinical tests performed by Matchett et al. in Syrian hamsters and K18-hACE2 mice. The authors documented a rapid recall of CD8^+^ memory T-cell responses upon i.n. viral challenge [61].

As an alternative tool for delivering the N protein into the organism, a VSV vector carrying S- and N-expressing constructs were used. Intranasal preimmunization of Syrian hamsters with this vaccine provided protection against challenges with several SARS-CoV-2 VOCs, whereas the i.m. administration was ineffective [62].

Routhu et al. created a complex vaccine based on the MVA vector containing the spike gene with a defective furin cleavage site and the gene encoding the N protein. When injected i.m. (this route appeared to be most effective), this vaccine candidate promoted the generation of serum and mucosal anti-spike IgG which were able to neutralize different VOCs and stimulated the expansion of both spike and N-specific CD4^+^ and CD8^+^ T cells. Vaccination prevented body weight loss and reduced viral load in primates challenged with Delta VOC [63].

Trivalent COVID-19 vaccine candidates bearing S-1, N and RdRp genes were proposed on the basis of human or chimpanzee adenoviral vectors. When administered once i.n. in mice, they were able to induce local and systemic specific humoral responses and generation of mucosal tissue-resident memory T (T_RM_) cells, as well as provide protection against the evolutionary distant B.1, B.1.1.7 and B.1.351 VOCs [64].

A vaccine candidate based on the mix of replication-deficient parapox viruses bearing N or S genes was named Prime-2-CoV_Beta. Like the same vector containing only the S gene, the combined prototype elicited significant amounts of neutralizing anti-S IgG and reduced lesions and viral loads in the lungs and the upper respiratory tract in challenged hamsters. The NHP challenge study revealed the ability of vaccines to provide rapid and long-lasting protection against SARS-CoV_2 infection [65].

Some attempts have been made to develop a combined vaccine based on the live attenuated influenza virus (LAIV) vector encoding N fragments in the influenza ORFs. For viral vectors bearing the N-expressing cassettes in NS [66] and NA [67] genes, high immunogenicity, i.e., triggering of cellular responses, and ability to confer protection against both COVID-19 and influenza infection, have been demonstrated. Therefore, this strategy seems to be promising for the coprophylaxis of several respiratory viral infections.

Table 1 summarizes the most significant findings from the preclinical studies of N-based prototypes of universal COVID-19 vaccine based on different expression platforms.

To date, only one N-based vaccine has been licensed for use in humans. Namely, the anti-coronavirus vaccine Convacell, which has proven to be an effective tool for the prevention of COVID-19 caused by the majority of SARS-CoV-2 strains, including BA.2.75 subvariant of Omicron VOC, has been authorized for clinical use in Russia [54]. Immunogenicity studies of the Convacell vaccine revealed comparable reactivity of PBMCs from vaccinated volunteers and COVID-19 convalescents in response to the N protein stimulation. Published flow cytometry results demonstrated similar increases in numbers of IFNγ- and IL-2-secreting CD4^+^ and CD8^+^ T cells in response to stimulation with the N proteins of Wuhan, Delta and Omicron lineages [72].

Another screening of individuals vaccinated with inactivated or mRNA vaccines revealed that the highest levels of anti-S and anti-N antibodies in the case of combined vaccination prevented both viral entry and intracellular replication [73]. The cellular immune responses in individuals with vaccine breakthrough infection target only the spike antigen, while the number of CD4^+^ T cells specific to the N protein is significantly higher in individuals with unvaccinated control infections [74].

In addition, some of the proposed prototypes have been studied in clinical trials (Table 2). Unfortunately, there are very limited published data evaluating the efficacy and safety of N-based COVID-19 vaccine candidates.

## 4. Mutability of N Sequences and Its Implication for the Performance of N-Based Diagnostics and Vaccines

Despite the fact that N-based vaccines are intended to be universal, even in the three years of SARS-CoV-2 circulation in the human population, minor changes appeared in this antigen. Figure 2 and Appendix A show the known immunogenic T-cell and B-cell N epitopes that are potentially affected by these mutations, respectively.

The substitutions in the N gene are not uniformly distributed, and apparently one of the most conserved regions is located in the central disordered linker proximal to the G215C mutation [75]. Thus, this conserved region was proposed to be used as a basis for universal vaccines [76].

The most frequently observed N mutations include R203K/G204R, P13L, S188L, S202N, D103Y, I292T, S194L, S197L, T339I, T148I and P344S [16,77,78]. Changes in phosphorylation sites are thought to have the greatest impact on the viral life cycle [79]. Another evolutionary trend typical for the N proteins of SARS-CoV and MERS-CoV is the mutational increase in the positive charge of the NLS (nuclear localization signal), which may lead to increased pathogenicity of SARS-CoV-2 due to enhanced interaction with host proteins [80]. However, future studies are needed to clarify the exact biological implications of the mentioned substitutions.

**Figure 2 vaccines-11-01810-f002:**
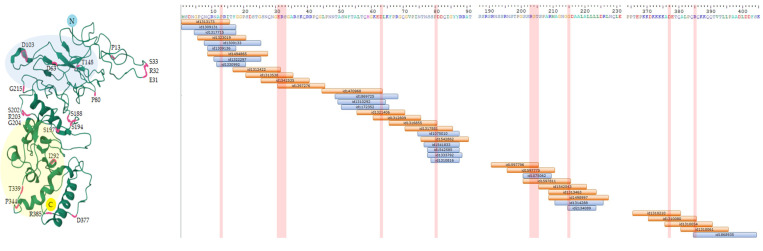
Reported mutations in the N protein of SARS-CoV-2 variants. (**a**) The most frequent mutations according to [16,31,77,78] mapped on the structure of a nucleoprotein dimer (PDB ID 8FG2); blue and yellow circles correspond to NTD and CTD, respectively; (**b**) CD4^+^ (orange) and CD8^+^ (blue) T-cell epitopes of the N protein of SARS-CoV-2 strain B.1 (Wuhan) deposited in the Immune Epitope Database, which contain variable residues uniquely distinguishing different VOCs (shown as red stripes). The alignment was visualized using Geneious 10.2.5 Software.

The R203K/G204R combination is considered the most important escape mutation in the N protein and was predominantly detected in 2019–2020 [81]. The highly phosphorylated serine-rich region, in which R203K/G204R mutations are localized, may recruit the host RNA helicase DDX1, contributing to template replication [43]. More recent studies involving a combined in silico and large-scale phylogenetic approaches revealed its adaptive nature, association with increased transmissibility of the B.1.1.7 lineage, COVID-19 severity [82] and worse clinical outcomes [83], probably due to the modulated ribonucleocapsid assembly. It was then proposed that this tandem mutation contributes to enhanced viral replication via the impairment of N phosphorylation by GSK-3 kinase, and the tendency to ablate the ancestral RG motif is evolutionarily advantageous [84]. Recombinant virus-like particles containing RG203/204KR mutations in the N protein-induced enhanced humoral immune responses and the formation of highly neutralizing serum IgG in mice [85]. Another variant of the detected substitution in position 203 of the N protein is R/M, which may alter the properties of peptides revealed to bind antiviral antibodies [86].

A docking study [87] revealed a potential site for targeting antiviral drugs (66–134) in the RNA-binding domain of the N protein, so escape substitutions such as P80R identified in the N protein of SARS-CoV-2 (P.1) may provide drug resistance to the mutated viruses.

The N proteins of B.1.1529 and B.1.351 VOCs containing the P13L/S mutation, which is localized in an epitope immunogenic for CD8^+^ T_RM_ and central memory T cells [88], may serve as a source of peptides identified as complexing with MHC class II [89]. Thus, this substitution may be considered as a result of the virus’ attempt to avoid the host humoral responses in COVID-19 [90].

Despite the slow evolutionary changes that may affect the antigenic properties of the N protein, the majority of studies on the development of N-targeting diagnostics suggest that serum or plasma N levels may be used to detect SARS-CoV-2 and predict the COVID-19 outcome, and the specificity of such antigenic tests may reach 100% [91]. Thus, the ACTIV-3/TICO Group study revealed a strong positive correlation of pulmonary disease severity or exacerbation risks with plasma N levels, as well as an association of this indicator with the lack of anti-S antibodies, male sex and renal failure [92]. Similarly, the feasibility of using the N antigen as a prognostic marker was reported by Wang et al., who identified 91.9% sensitivity and 94.2% specificity of the N-targeting test system. The higher the plasma N concentration was, the more intensive therapy was needed, but the rate of decline in antigen levels was not associated with disease severity [93]. Another N-based fluorescence immunochromatographic test system demonstrated 75.6% sensitivity and 100% specificity when tested in 251 participants and thus can be recommended for early SARS-CoV-2 detection [94]. At the same time, some N mutations, in particular T135I in the B.1.1.7 variant, have been reported to pose a significant diagnostic risk, as they allow the N antigen to avoid recognition by diagnostic antibodies [95].

The persistence rate of anti-N antibodies differs from that of broadly accepted diagnostics anti-S IgG [96], but they may also be used qualitatively as a hallmark of infection and quantitatively as a prognostic marker of its outcomes. ELISA screening of donor sera on the captured N protein has shown the possibility of detecting anti-N antibodies regardless of the VOC that caused the infection due to high cross-reactivity [18], and such diagnostic tests are characterized by high sensitivity and specificity [97]. At the same time, the role of anti-N antibodies in providing protection against COVID-19 remains unclear. Some studies have shown that mice passively immunized with N-specific sera or mAbs are resistant to the development of severe pathology, probably due to the stimulation of NK-mediated ADCC reactions [98].

## 5. Concluding Remarks

T-cell responses represent a major immune component involved in protection against COVID-19 and other respiratory infections [99,100], so the development of new vaccines based on viral antigens that stimulate this mode of immunity seems to be feasible. Indeed, given its highly conserved nature, immunogenicity and the abundance of the N protein in infected cells, it is difficult to consider another viral antigen as a more suitable target for the development of universal COVID-19 vaccines. Our mini-review summarizes the results of preclinical evaluation of universal N-based vaccine prototypes developed on the basis of different platforms, including viral vectors, recombinant antigens formulated with different adjuvants or mRNA vaccines, which aim at providing protection against different SARS-CoV-2 VOCs. Comprehensive further studies are needed to clarify the optimal route of administration and the most appropriate adjuvant to ensure prolonged N persistence and to stimulate immune responses of a certain type. The sufficiency of administering N protein alone or using it in combination with other viral antigens remains a subject of future studies.

Despite the fact that the N protein is also susceptible to evolutionary changes, they are of the slowest nature, and conserved regions similar among VOCs are found in the N molecule. For example, it has been demonstrated that anti-N serology tests based on the ancestral antigen are still able to detect antibody responses to recent SARS-CoV-2 variants [18]. However, studies have revealed the impact of rare mutations on N antigenicity and immunogenicity [31], which may potentially alter the performance of N-based test systems and vaccine candidates; so, N protein variability should be taken into account as a cause of potential antigenic mismatch between the epitope composition of an N-based vaccine and the circulating SARS-CoV-2 strain. This limitation is fair considering that a single amino acid substitution in the T-cell epitope is sufficient to affect the peptide immunogenicity [101]. Side-by-side studies are needed to compare the efficacy of vaccine prototypes based on the ancestral N protein or N antigen of evolutionarily new strains.

An additional attractiveness of the N protein for the development of broad-spectrum vaccines is related to its ability to trigger not only cellular but also humoral immune responses. From this point of view, abundantly produced anti-N antibodies, lacking neutralizing activity, may mediate Fc-engaging innate cytolytic reactions. However, slow evolutionary changes in the N sequence may also be responsible for the reduced recognition of anti-N antibodies generated as a result of immunization with a previously developed vaccine of the N antigen of the current SARS-CoV-2 strain. Given the moderate mutability of the N protein, targeting the occasionally updated antigen alone or in combination with other viral proteins seems to be the optimal strategy for further vaccine development.

Despite the obvious advantages of N-based vaccines in terms of a broader spectrum of inducible immune responses and less dependence on the evolutionary variability of the virus, the development and practical implementation of such vaccines face a number of challenges. The major difficulty is the lack of established correlates of protection for COVID-19 T-cell vaccines, i.e., there is no clear data on what levels of particular T-cell subsets will be sufficient to accelerate pathogen elimination from the infected organism, thereby, promoting recovery. In addition, evaluation of vaccine candidates aimed at inducing a T-cell response is hampered due to the fact that specific humanized animal models are required for the correct presentation of human T-cell epitopes, and even such humanized animals may not be suitable for some vaccines based on viral vectors due to the immunodominance phenomenon [67]. Furthermore, direct comparison of the performance of different T-cell-based vaccines is challenging due to the lack of standardized assays for measurement of virus/epitope-specific T-cell responses. Finally, an optimal platform for the development of N-based vaccines is still a subject of discussion, as the most suitable adjuvant for recombinant N protein formulation is unknown, and numerous difficulties accompany the design and implementation of vaccine candidates involving delivery systems such as nanoparticles and viral vectors. In general, all these challenges have been recognized in the development of broadly reactive influenza T-cell vaccines [102], and more research will be needed to harmonize assays and standards for assessing the efficacy of N-based COVID-19 vaccines.

## Figures and Tables

**Figure 1 vaccines-11-01810-f001:**
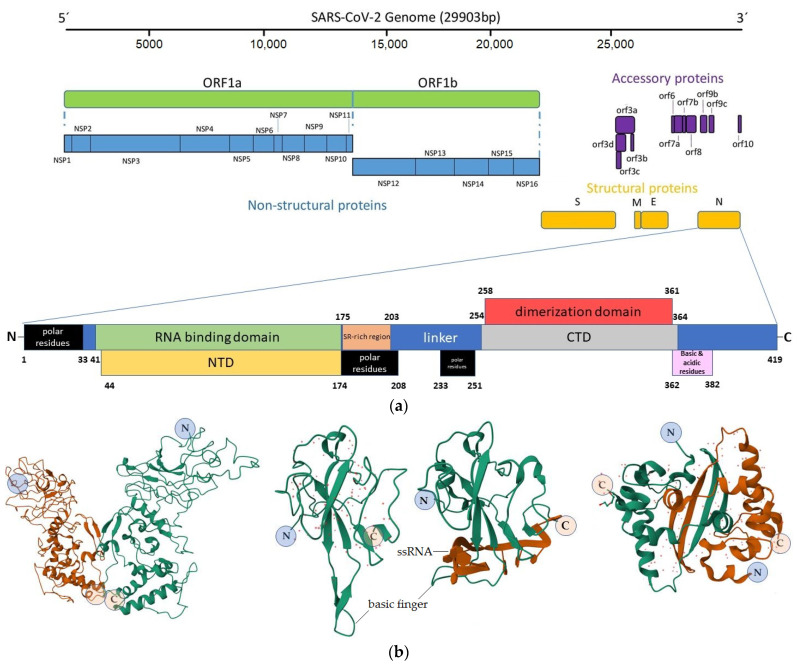
Structure of the SARS-CoV-2 N protein. (**a**) The scheme of N molecule domains involved in the RNA binding and other interactions; adapted from [30,31]. Numbers indicate the position of N amino acid residues; (**b**) side view of the N protein dimer (PDB ID 8FG2), crystal structure of the NTD (PDB ID 7VNU), NTD complexed with ssRNA (PDB ID 7ACT) and a crystal structure of the CTD dimer (PDB ID 7C22). N and C indicate the N- and C-terminal part of the molecules, respectively.

**Table 1 vaccines-11-01810-t001:** Some examples of N-based COVID-19 vaccine candidates.

Carrier and Antigen	Animals	Immunization Route	Main Results	Ref.
Recombinant N and RBD proteins	Mice	s.c.	Triple immunization of mice with 15 µg of *E. coli*-derived *E. coli* proteins supplemented with saponin led to the generation of neutralizing anti-RBD and anti-N IgG and stimulated the expansion of RBD/N-specific CD4^+^/CD8^+^ T cells.	[47]
Recombinant N protein	Rats	i.m.	Wistar and Lewis rats were immunized with the N protein of bacterial origin at a dose of 150 µg 4 times, 7 days apart. The vaccine appeared to be safe, non-pyrogenic and induced the generation of N-specific IgG as well as activation of lung CD68^+^ macrophages in Lewis rats.	[68]
Recombinant N protein	Hamsters	i.m.	The vaccine OVX033 represented the product of the fusion of the heptamer-forming OVX313 domain and the full-length N protein. Hamsters were immunized twice, 28 days apart, with 50 µg of OVX033 with or without SQ adjuvant. Regardless of adjuvant use, the vaccine provided protection against the challenge with 10^4^ TCID_50_ of SARS-CoV-2 B.1, B.1.617.2 or B.1.1.529, and induced N-specific humoral and cross-reactive cellular responses in lungs and spleen.	[50]
Recombinant N protein or its fragments	Mice	i.p.	Mice were immunized 3 or 4 times at 2-week intervals with N-protein (75 µg) or N-derived peptide (20 µg) in complete (for prime vaccination) or incomplete Freund’s adjuvant (for booster doses). Long-lasting (up to 22 weeks) anti-N IgG titers and increased numbers of CD4^+^ and CD8^+^ memory T cells were revealed. Moreover, full-length N protein induced the secretion of pro-inflammatory and Th1-associated cytokines (TNFα, GCSF, IFNγ, CCL2, CCL5, etc.)	[48]
Recombinant N-derived peptides	Mice	i.p.	The vaccine prototype included the immunogenic peptides from S2 and N antigens adsorbed on an aluminum hydroxide gel with or without the addition of MPLA. The mice immunized with 25 µg of each peptide twice at 15-day intervals exhibited high-titer anti-P1, anti-P2 and anti-N IgG responses, generation of neutralizing IgG and increased antigen-specific levels of INFγ in the splenocytes.	[51]
Recombinant N protein	Mice	s.c.	Mice were primed with 100 µg of the N protein in complete Freund’s adjuvant and boosted 2 and 4 weeks later with 50 µg of the N protein in incomplete Freund’s adjuvant. The vaccine which was recognized by patients’ sera, induced robust serum anti-N IgG, IgM and INFγ responses.	[52]
Recombinant N protein of Wuhan-Hu-1	Mice	?	Three different adjuvants (Alum, AS03 and Montanide/ISA720) were tested as supplements for 40, 80 or 120 µg of recombinant RBD, SS1 fragment or SS1 + N/SS1 + RBD cocktail. Animals immunized on days 0, 21 and 35 developed high titers of specific IgG and high levels of IL-12, IFNγ and IL-4 in serum, but only the Alum group had significantly increased levels of neutralizing antibodies. Immunized primates exhibited a decrease in viral loads 2–7 days after i.n. challenge with 10^6^ homologous SARS-CoV-2. The vaccine safety was demonstrated by LAL testing and histopathologic studies.	[53]
Recombinant N protein	Mice	i.m.	The vaccine composed of the recombinant N protein was mixed with a squalane-based emulsion (1:1 *v*/*v*). The mice were immunized with 50 µg of vaccine twice, 14 days apart. The vaccine prototype induced the long-lasting (up to 386 days) anti-N IgG and specific T-cell responses with a mixed Th1/Th2 phenotype (IFNγ, IP-10, IL-4 and IL-5 secreting).	[54]
	Hamsters		Animals received two doses (50 μg) of vaccine 2 weeks apart and were challenged with 10^5^ TCID_50_ of SARS-CoV-2 (B.1.1) 2 weeks after the last injection. The titers of N-specific IgG peaked on days 21–28 after the first immunization and were detectable even on 386-day p.i. In challenge study, weight loss, lung lesions and viral loads were significantly reduced in immunized hamsters.	
	NHPs		The marmosets were immunized with 50 µg of vaccine twice at a 21-day interval. The vaccine was capable of inducing N-specific CD3^+^, CD4^+^ and CD8^+^ T-cell responses.	
Recombinant N-containing fusion protein	Mice	i.n.	18 µg of vaccine based on RBD fused with NTD of the N protein was proposed to boost mice previously inoculated twice i.m. with S-expressing mRNA vaccine. Vaccination resulted in dramatically enhanced generation of virus-neutralizing mucosal IgA and serum IgG antibodies.	[69]
Recombinant N protein of Wuhan-Hu-1	Mice	i.m.	RelCoVax vaccine, consisting of recombinant RBD and N antigens formulated with Alum or CpG, was administered twice at 2-week intervals at doses of 1–4 µg (priming) and 10 µg (boosting). The vaccination evoked dose-dependent virus-neutralizing and anti-N/RBD humoral responses and generation of IFNγ-producing antigen-specific splenocytes.	[70]
Virus-like particles containing N of Wuhan-Hu-1	Mice	s.c.	Virus-like particles (VLPs) expressed in HEK293 cells and similar in shape and size to the SARS-CoV-2 virions were proposed as carriers of S, N, M and E antigens. They were adsorbed onto 2% Alhydrogel and adjuvanted with K3-CpG ODN to vaccinate mice twice 14 days apart with 0.75–24 µg of VLPs per animal. The vaccine prototype was able to provoke potent cross-specific humoral and CD4^+^ T-cell responses. The challenge study in K18 hACE2 transgenic mice was started on 36th day after the last immunization by i.n. infection with 10^5^ PFU of hCoV-19/Turkey/Pen07/2020 (B.1.1) virus. In mice previously vaccinated twice with 8 µg of VLPs, protection manifested as a reduction in acute lung injury and viral loads.	[55]
	Rats		Virus-like particles mimicking SARS-CoV-2 virions mixed with 2% Alhydrogel and K3-CpG ODN adjuvant were administered twice with 2-week intervals at a dose of 40 µg. High virus-neutralizing titers were observed 2 weeks after boosting.	
	Ferrets		Ferrets received two doses (10 or 40 µg) of virus-like particles, adjuvanted with 2% Alhydrogel and K3-CpG ODN, twice (2 weeks apart). Robust dose-independent generation of virus-neutralizing antibodies was detected 2 weeks after priming and boosting.	
Nanoparticles bearing the recombinant N protein	Mice	s.c.	The vaccine comprised acetylated dextran-based nanoparticles bearing the N protein, which, when injected three times (14 days apart) at a dose of 50 µg, induced much stronger spleen CTL and serum humoral responses compared with monomeric N.	[56]
	Rabbits		Triple immunization with 50 µg of N-bearing nanoparticles at 2-week intervals induced more intense generation of N-specific IgG compared with the monomeric N administration.	
DNA encoding N protein of huCoV-19/WH01	Mice	i.m.	The combined expression construct consisted of RBD fragments from the huCoV-19/WH01, Alpha and Beta VOCs, M and N genes. When injected in mice at a dose of 50 μg three times at 2-week intervals, it promoted the generation of cross-neutralizing anti-S antibodies and elicited N-specific IFNγ^+^ T-cell responses. A challenge study performed in a K18-hACE2 mouse model two weeks after the last immunization (1 × 10^5^ PFU SARS-CoV-2 of B.1.351 strain) revealed complete protection by viral load reducing and lung tissue protection.	[57]
	Rabbits		Rabbits were immunized 4 times with 84 or 840 μg of DNA-based vaccine containing the N-cassette. The formation of N-specific and anti-S neutralizing IgG antibodies, as well as the induction of cross-reactive N(Bat-CoV)-specific IFNγ-secreting T cells were detected for both dosing regimens on 15-day p.i.	
mRNA expressing N protein of Wuhan-Hu-1	Mice	i.m.	Combined vaccination with mRNAs expressing N and prefusion stabilized S protein (1 μg for each, twice 3 weeks apart) led to intense appearance of antigen-specific IgG, IFNγ/TNFα/IL-2-secreting CD4^+^/CD8^+^ effector memory T cells. The mRNA-N vaccination alone moderately reduced the lung viral titer of challenge virus MA-SARS-CoV-2 CMA4 (2 × 10^4^ PFU), but the complex vaccine conferred more effective antiviral protection.	[58]
	Hamsters		The mRNA-based vaccines expressing N and/or S antigens were administered at a dose of 2 μg each twice, 3 weeks apart. Two weeks after the booster vaccination, challenge with 2 × 10^4^ PFU of SARS-CoV-2 B.1.617.2 or B.1.1.529 variants was performed. Viral titers in the lungs and nasal tissues, lung lesions and weight loss were significantly reduced in hamsters vaccinated with mRNA-S, mRNA-N alone (even in the absence of neutralizing antibodies) or mRNA-N+S.	
mRNA expressing N epitopes	Mice	i.m.	LNP-formulated mRNA vaccines expressing S antigens of Delta/Omicron VOCs or a cassette consisting of conserved epitopes (including the N sequence) were administrated at a dose of 10 µg twice, 21 days apart. In all cases, the vaccines elicited robust S-specific IgG responses, including neutralizing and generation of effector memory T cells. Two weeks after boosting, mice were challenged i.n. with 5 × 10^3^ TCID_50_ of SARS-CoV-2 B.1.617.2 variant or with 5 × 10^3^ PFU of SARS-CoV-2 MA30. Immunization with complex vaccine resulted in maximal reduction of morbidity and mortality, complete absence of viral antigens in lung tissue and zero lung viral load.	[71]
Self-amplifying RNA expressing N protein of Wuhan-Hu-1	Mice	i.m.	The saRNAs expressing RBD and N antigens were LNP encapsulated and administered in a prime-boost regimen at 21-day intervals at a dose of 0.5 μg each. Vaccination induced high titers of neutralizing antibodies against B.1.351 and B.1.617.2 SARS-CoV-2 variants and expanded both S- and N-specific IFNγ/TNFα/IL-2/-secreting (Th1-biased) CD3^+^ CD4^+^/CD8^+^ T cells.	[59]
	Hamsters		Vaccination with a cocktail of RBD- and N-expressing saRNAs was performed at a dose of 1 or 5 μg (twice, 21 days apart). Two weeks after booster vaccination, hamsters were i.n. challenged with 1 × 10^4^ TCID_50_ of WA1/2020 SARS-CoV-2 virus, and lung viral loads as well as lung tissue pathology were dose dependently reduced in immunized animals.	
Ad5-vector expressing N gene	Mice	i.m.	K18-hACE2 mice were immunized with 10^9^ PFU of vaccine vectors bearing S and/or N genes and then challenged i.n. with 5 × 10^4^ PFU of SARS-CoV-2 (USA-WA1/2020). High titers of specific IgG and robust CD4^+^/CD8^+^ responses were revealed 3 weeks after immunization. The N-based vaccine alone did not confer acute lung protection, but in combination with the S-based vector provided protection of the brain.	[60]
VSV vector expressing N gene	Hamsters	i.n., i.m.	Hamsters were vaccinated with 1 × 10^5^ PFU of mixed vectors expressing S or N gene and then challenged with 1 × 10^5^ TCID_50_ of nCoV-WA1-2020, B.1.1.7, B.1.351 and P.1 VOCs. The i.n. vaccination diminished viral loads in oral swabs and in the lungs, and caused no or minimal interstitial pneumonia. Only i.n.-vaccinated hamsters generated a sustained humoral response to either S or N antigen.	[62]
Ad5-vector expressing N gene	Mice	i.v.	The K18.hACE2 mice were immunized with 5 × 10^10^ particles of N-expressing vector and then i.n. challenged with 3 × 10^2^ PFU of nCoV/USA_WA1/2020 (WA) virus. Ad5-N vaccine reduced the challenge-caused lung viral load, weight loss and mortality, as well as established N-specific memory CD8^+^ T cells, which migrated to the site of viral challenge.	[61]
	Hamsters		Inoculation with 1.8 × 10^11^ particles of recombinant N-bearing adenovirus followed by challenge with 6.75 × 10^5^ PFU of nCoV/USA_WA1/2020 (WA) or SARS-CoV-2 (B.1.1.7) resulted in a significant decrease in median lung viral titers.	
MVA-vector expressing N gene	Mice	i.m.	The i.m. injection of 10^7^ PFU of MVA-vector expressed N and S protein with an inactivated furin cleavage site was performed two times at a 4-week interval. Vaccination prevented body weight loss and reduced lung viral load upon challenge with 10^5^ PFU of WA-1/2020 virus or SARS-CoV-2 (B.1.351).	[63]
	NHPs	i.m., s.l., buccal	Rhesus macaques received 10^8^ PFU of a recombinant MVA-based vector expressing N and S (non-cleavable) antigens twice at a 4-week interval. This vaccination strategy promoted the production of serum or mucosal (depending on the administration route) cross-neutralizing anti-S IgG and S/N specific CD4^+^ and CD8^+^ T cells 2 weeks after the second immunization. Vaccination also reduced viral titers, lung pathology score, mortality and morbidity upon subsequent i.n. or i.t. challenge with 10^5^ PFU of B.1.617.2 variant.	
Ad5- and Ad68-vectors expressing N gene of Wuhan-Hu-1	Mice	i.n., i.m.	Recombinant trivalent vaccine vectors developed on the basis of replication-deficient human or chimpanzee adenoviruses expressing S-1, N and RdRp genes were proposed for single i.n. administration at a dose of 5 × 10^7^ PFU. Intranasal injection induced mucosal and serum-specific humoral responses and T_RM_ cell generation in the upper airways, as well as provided protection (i.e., reduced viral loads and weight loss) against lethal doses (1 × 10^5^ PFU) of B.1, B.1.1.7 and B.1.351 SARS-CoV-2 lineages.	[64]
Parapox virus-based vector expressing N gene	Mice	i.m.	The vaccine containing a mix of replication-deficient parapox viruses bearing N or S expression constructs was administered twice with 21-day interval at a dose of 10^7^ PFU. It induced robust and prolonged antigen-specific Th1-biased humoral and CD4^+^/CD8^+^ T-cell responses.	[65]
	Hamsters		Animals were immunized twice (28 days apart) with 10^6^–10^8^ PFU of a cocktail of recombinant N or S-expressing parapox-based viruses. Challenge with 10^2^ TCID_50_ of SARS-CoV-2 (B.1) was performed on day 28 after the second immunization, and vaccinated animals experienced a dramatic decrease in lung viral titers and minimally affected upper and lower airways.	
	NHPs		Rhesus macaques received two doses (10^6^–10^8^ PFU) of mixed parapox-based viruses carrying N or S genes, 28 days apart. Vaccination induced the generation of virus-neutralizing anti-S antibodies and IFNγ-secreting T cells. Viral titers in nasal turbinates and bronchoalveolar lavage were reduced several fold in vaccinated animals upon subsequent i.n./i.t. challenge with 10^5^ TCID_50_ of SARS-CoV-2 (B.1).	
Influenza virus vector (PR8 backbone) expressing N fragments	Mice	i.n.	N protein fragments (231–309 or 211–369) were inserted into the ORF of the truncated influenza NS1 protein of H1N1 and H3N2 PR8-based reassortants. Mice received 6.0 lgEID_50_ of recombinant viruses twice (3 weeks apart) or were i.m. primed with inactivated SARS-CoV-2 and then boosted once with N-bearing vectors. After vaccination, expansion of N-specific IFNγ/TNFα/IL-2-producing CD8^+^ T cells was detected. Mice were challenged with 10 MLD_50_ of different influenza viruses or with SARS-CoV-2 (B.1.1.351); weight loss and viral loads were reduced in all immunized groups, especially when a prime-boost immunization strategy was implemented.	[66]
	Hamsters	i.n.	Immunization with recombinant influenza virus containing the N fragment (211–369) was performed once at a dose of 7.0 lgEID_50_. After 4 weeks, hamsters were subjected to i.n. challenge with 5.0 lgTCID_50_ of SARS-CoV-2 (B.1.1), and viral titers were significantly lower in samples from the immunized group relative to the control animals.	[66]
Influenza virus vector (Len/17 backbone) expressing N fragments	Hamsters	i.n.	Hamsters were inoculated with two doses (21 days apart) of 5 × 10^6^ EID_50_ of the recombinant influenza virus vector bearing polyepitope N regions 92–118 and 293–370 in the ORF of NA. Vaccination evoked the influenza-specific immune responses and protected animals against infection with 10^6^ EID_50_ of Sh/PR8 influenza virus or with 10^5^ TCID_50_ of SARS-CoV-2 (B.1/B.1.617.2). Reduced weight loss and clinical manifestation of SARS-CoV-2-induced pneumonia correlated with the induction of IFNγ-secreting T-cell responses.	[67]

NHPs: non-human primates; CTLs: cytotoxic T lymphocytes; Ad5: adenovirus type 5; VSV: vesicular stomatitis virus; PFU: plaque forming unit; MVA: modified vaccinia virus Ankara; TCID_50_: median tissue culture infectious dose; VLPs: virus-like particles; IgG: immunoglobulin class G; IgA: immunoglobulin class A; EM: effector memory; IFNγ: interferon gamma; IL-2: interleukin-2; IL-4: interleukin-4; TRM: tissue-resident memory; TNFα: tumor necrosis factor alpha; ADCC: antibody-dependent cellular cytotoxicity; mRNA: matrix ribonucleic acid.

**Table 2 vaccines-11-01810-t002:** Clinical trials of N-based vaccines against COVID-19.

Company	Phase	Year	Clinicaltrials.Dov Identifier	Available Results
Speransa Therapeutics (Frankfurt am Main, Germany)	I	2022–2023	NCT05367843	No
University Hospital Tübingen (Tübingen, Germany)	I	2022–2023	NCT05389319	No
St. Petersburg Research Institute of Vaccines and Sera (St. Petersburg, Russia)	I/II	2021–2022	NCT05156723	Yes [72]
St. Petersburg Research Institute of Vaccines and Sera (St. Petersburg, Russia)	II/III	2022–2023	NCT05726084	No
Research Institute of Influenza (St. Petersburg, Russia)	I/II	2022–2023	NCT05696067	No
The Scientific and Technological Research Council of Turkey (Ankara, Turkey)	I	2021	NCT04818281	No
The Scientific and Technological Research Council of Turkey (Ankara, Turkey)	II	2021–2022	NCT04962893	No
ImmunityBio, Inc. (El Segundo, CA, USA)	I	2022	NCT04843722	Study withdrawn

## Data Availability

Data sharing not applicable.

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
