# Peer review of "Overview of Nucleocapsid-Targeting Vaccines against COVID-19"

_vaccines, 2023, doi:10.3390/vaccines11121810_

Round 1

Reviewer 1 Report

Comments and Suggestions for Authors

The abstract of the review manuscript provides a comprehensive overview of the challenges associated with developing vaccines against SARS-CoV-2, particularly focusing on the nucleocapsid protein (N) as a potential target. However, it lacks specific details about the methods, data sources, and the criteria used for assessing the efficacy of N-based vaccines in animal models or clinical trials, making it challenging for readers to evaluate the quality of evidence presented. Furthermore, the abstract mentions the accumulation of escape mutations in the N protein but does not delve into the potential consequences or implications for vaccine development in sufficient depth. Additionally, it does not discuss the broader context of COVID-19 vaccine development, such as the success of spike protein-based vaccines, which could provide a more balanced perspective on the subject.

The introduction section of this review manuscript provides a comprehensive overview of the challenges and potential benefits associated with using the nucleocapsid (N) protein of SARS-CoV-2 as a target for COVID-19 vaccines. However, it opens with a somewhat misleading statement suggesting that the pandemic is finished, which may not be accurate depending on the timing of the manuscript. Additionally, while it highlights the importance of N-based vaccines, it lacks a clear statement of the research question or objective of the review, making it challenging for readers to discern the specific focus and purpose of the manuscript. Furthermore, the introduction could benefit from a more concise and structured presentation of key points and a clearer connection between the discussed topics, such as the role of N protein, the challenges of spike protein-based vaccines, and the need for N-based vaccines.

The subsection on the "SARS-CoV-2 nucleoprotein structure and functions" provides a detailed overview of the N protein's properties and functions, but the organization of information appears somewhat disjointed, making it challenging for readers to follow the logical flow of the content. The information on the N protein's modular organization and its functions in genome packaging is presented in a scattered manner, lacking a clear structure that would facilitate comprehension. Additionally, while the subsection mentions various domains and motifs involved in RNA binding and polymerization, it could benefit from more visual aids or diagrams to help illustrate these complex molecular interactions. The subsections “Development of N-based broadly protective vaccine prototypes” and “Mutability of N sequences and its implication for the performance of N-based diagnostics and vaccines” are well written and elaborated as doesn’t require modifications.

The conclusion section of this manuscript has several shortcomings. It lacks a clear restatement of the research objective, making it challenging for readers to discern the primary focus of the review. Additionally, it offers a rather abrupt ending without providing a comprehensive synthesis of the reviewed information or overarching insights, leaving the reader without a clear takeaway. While it introduces the concept of slow evolutionary changes in the N protein, it does not elaborate on the potential consequences for vaccine development or provide practical recommendations. There is also a notable absence of a discussion on the review's limitations and areas for future research, which are customary components of scientific conclusions.

Comments on the Quality of English Language

Minor editing of the English language is required.

Author Response

Comments from Referee 1:

The abstract of the review manuscript provides a comprehensive overview of the challenges associated with developing vaccines against SARS-CoV-2, particularly focusing on the nucleocapsid protein (N) as a potential target. However, it lacks specific details about the methods, data sources, and the criteria used for assessing the efficacy of N-based vaccines in animal models or clinical trials, making it challenging for readers to evaluate the quality of evidence presented. Furthermore, the abstract mentions the accumulation of escape mutations in the N protein but does not delve into the potential consequences or implications for vaccine development in sufficient depth. Additionally, it does not discuss the broader context of COVID-19 vaccine development, such as the success of spike protein-based vaccines, which could provide a more balanced perspective on the subject.

Authors’ response: We thank the reviewer for this note. Although the number of words in the annotation is very limited, we have added the data on the major methods and criteria used in testing of the effectiveness of N-based vaccines. We also indicated the possibility of the influence of escape mutations accumulated in this antigen on N protective properties. In addition, the beginning of the abstract has been rephrased to mention the success of the development of Spike-based vaccines against COVID-19.

The introduction section of this review manuscript provides a comprehensive overview of the challenges and potential benefits associated with using the nucleocapsid (N) protein of SARS-CoV-2 as a target for COVID-19 vaccines. However, it opens with a somewhat misleading statement suggesting that the pandemic is finished, which may not be accurate depending on the timing of the manuscript. Additionally, while it highlights the importance of N-based vaccines, it lacks a clear statement of the research question or objective of the review, making it challenging for readers to discern the specific focus and purpose of the manuscript. Furthermore, the introduction could benefit from a more concise and structured presentation of key points and a clearer connection between the discussed topics, such as the role of N protein, the challenges of spike protein-based vaccines, and the need for N-based vaccines.

Authors’ response: We agree with the reviewer that SARS-CoV-2 remains a major cause of the incidence of respiratory infections, but here we meant that the official end of the pandemic was declared by WHO in May 2023. Therefore, we thought it would be appropriate to talk about this in our manuscript. According to the comments, we outlined in the introduction the objective of our review and more clearly reflected the logical relationship of the topics discussed in the paper.

The subsection on the "SARS-CoV-2 nucleoprotein structure and functions" provides a detailed overview of the N protein's properties and functions, but the organization of information appears somewhat disjointed, making it challenging for readers to follow the logical flow of the content. The information on the N protein's modular organization and its functions in genome packaging is presented in a scattered manner, lacking a clear structure that would facilitate comprehension. Additionally, while the subsection mentions various domains and motifs involved in RNA binding and polymerization, it could benefit from more visual aids or diagrams to help illustrate these complex molecular interactions. The subsections “Development of N-based broadly protective vaccine prototypes” and “Mutability of N sequences and its implication for the performance of N-based diagnostics and vaccines” are well written and elaborated as doesn’t require modifications.

Authors’ response: We thank the reviewer for this valuable comment. To make information about the structure of the N protein easier to understand, we have provided a detailed scheme of the domain organization of its molecule. However, following these comments, we have added a model of the interaction of the N protein with the viral genome in the section “SARS-CoV-2 nucleoprotein structure and functions”. The logical structure of this part of the manuscript has also been modified. We greatly appreciate the positive feedback on the next subsections.

The conclusion section of this manuscript has several shortcomings. It lacks a clear restatement of the research objective, making it challenging for readers to discern the primary focus of the review. Additionally, it offers a rather abrupt ending without providing a comprehensive synthesis of the reviewed information or overarching insights, leaving the reader without a clear takeaway. While it introduces the concept of slow evolutionary changes in the N protein, it does not elaborate on the potential consequences for vaccine development or provide practical recommendations. There is also a notable absence of a discussion on the review's limitations and areas for future research, which are customary components of scientific conclusions.

Authors’ response: We agree with the reviewer that the Conclusion section was not clearly presented. The structure of the “Concluding Remarks” subsection has been modified. In particular, we have added a restatement of this review objective and summary conclusions, indicated possible prospects for future research, outlined limiting factors and made recommendations for the development of future N-based vaccines against COVID-19, given the slow evolutionary variability of this antigen. In addition, we have outlined the main challenges faced by the developers of such vaccines.

Minor editing of the English language is required.

Authors’ response: English editing was performed by a native speaker.

Reviewer 2 Report

Comments and Suggestions for Authors

Very well organized mini-review on nucleocapsid-targeting vaccines for COVID19. The authors have done a great job on highlighting various aspects of vaccine design, platforms and adjuvants with a focus on covid-19 nucleocapsid protein.

I would recommend this review for publication. 

Author Response

Comments from Referee 2:

Very well organized mini-review on nucleocapsid-targeting vaccines for COVID19. The authors have done a great job on highlighting various aspects of vaccine design, platforms and adjuvants with a focus on covid-19 nucleocapsid protein.

I would recommend this review for publication.

Authors’ response: We thank the reviewer for this positive feedback.

Reviewer 3 Report

Comments and Suggestions for Authors

The authors hypothesize that a vaccine directed against the nucleocapsid (N) protein of COVID-19 might be an effective means to prevent COVID-19 infection. Although there are details provided about the N protein, and its development as a potential vaccine, using hypothetical recombinant protein, adjuvants, nanoparticles, DNA or mRNA delivery systems, viral vectors, and potential combination with existing type mRNA systems, it is unlikely that any practical use would come of further development of potential N-protein vaccines, given the extensive research and successful development of existing mRNA vaccines. There would be issues using viral vector vaccines, and perhaps limited practicality of any nanoparticle vaccines.

Comments on the Quality of English Language

There is need for extensive editing of the quality of the English used in this manuscript.

Author Response

Comments from Referee 3:

The authors hypothesize that a vaccine directed against the nucleocapsid (N) protein of COVID-19 might be an effective means to prevent COVID-19 infection. Although there are details provided about the N protein, and its development as a potential vaccine, using hypothetical recombinant protein, adjuvants, nanoparticles, DNA or mRNA delivery systems, viral vectors, and potential combination with existing type mRNA systems, it is unlikely that any practical use would come of further development of potential N-protein vaccines, given the extensive research and successful development of existing mRNA vaccines. There would be issues using viral vector vaccines, and perhaps limited practicality of any nanoparticle vaccines.

Authors’ response: We thank the reviewer for this comment. In this work, we unbiasedly summarized previously published data on the effectiveness of N-targeting vaccines developed on the basis of various platforms. Despite some contradictory data or negative attempts, the results of most preclinical trials indicate the feasibility of developing cross-protective vaccines against COVID-19 based on the recombinant N protein or N-encoding tools and are in good agreement with theoretical ideas about its conservation. Thus, there is a large background for further practical implementation of these developments, but it is difficult to predict which strategy for creating N-based vaccines will be the most successful. In addition, we have outlined the main challenges faced by the developers of such vaccines in the “Concluding remarks” section.

There is need for extensive editing of the quality of the English used in this manuscript.

Authors’ response: English editing was performed by a native speaker.

Reviewer 4 Report

Comments and Suggestions for Authors

The authors give a detailed overview about the nucleocapsid-targeting COVID-19 vaccines. The review is detailed, understandable and refers to a lot of relevant publications. Please, review the reference list (e.g. reference 67: accepted manuscript, but the date is 2022 for this one, so this could be updated). Also, on page 2, row 72: antibody-mediated cellular phagocytosis/cytotoxicity, I know that this is used as well, but I would recommend to replace this to antibody-dependent cellular phagocytosis/cytotoxicity, given that this stands for the acronym.

I recommend the manuscript for publication in Vaccines.

Author Response

Comments from Referee 4:

The authors give a detailed overview about the nucleocapsid-targeting COVID-19 vaccines. The review is detailed, understandable and refers to a lot of relevant publications. Please, review the reference list (e.g. reference 67: accepted manuscript, but the date is 2022 for this one, so this could be updated). Also, on page 2, row 72: antibody-mediated cellular phagocytosis/cytotoxicity, I know that this is used as well, but I would recommend to replace this to antibody-dependent cellular phagocytosis/cytotoxicity, given that this stands for the acronym.

I recommend the manuscript for publication in Vaccines.

Authors’ response: We thank the reviewer for this positive feedback. The list of references was checked and corrected, as well as the indicated term.

Round 2

Reviewer 3 Report

Comments and Suggestions for Authors

Thanks to the authors for including the limitations to the proposed use of nucleocapsid-targeted vaccines against COVID-19. Similar limitations should be included regarding delivery systems, ie nanoparticles, viral particles.

Comments on the Quality of English Language

No comments

Author Response

Thanks to the authors for including the limitations to the proposed use of nucleocapsid-targeted vaccines against COVID-19. Similar limitations should be included regarding delivery systems, ie nanoparticles, viral particles.

Authors’ response: We thank the reviewer for the positive feedback and note. We have outlined the delivery factors complicating the development of N-based vaccines in the "Concluding remarks" section.